# Seed treatment using methyl jasmonate induces resistance to rice water weevil but reduces plant growth in rice

Emily C. Kraus *, Michael J. Stout

Louisiana State University AgCenter, Baton Rouge, Louisiana, United States of America

* krausec07@gmail.com

**Data Availability Statement:** All relevant data are within the manuscript and its Supporting Information files.

## Abstract

The jasmonic acid cascade plays a pivotal role in induced plant resistance to herbivores. There have been a number of investigations into the potential uses of derivatives of this hormone for pest management. Understanding the phenotypic plasticity of plant defense traits interactions in agricultural systems may facilitate the development of novel and improved management practices, which is desirable as management of insects in most agricultural systems is currently heavily reliant on insecticides. The rice water weevil (RWW), *Lissorhoptrus oryzophilus* Kuschel, is a pest of rice, *Oryza sativa*, in the southern U.S. and globally. The effects of the jasmonic acid derivative, methyl jasmonate (MJ), on induced defenses to RWW in rice, and the potential costs of MJ-induced resistance to plant growth and fitness, were tested in a series of field and greenhouse trials. It was hypothesized that seed treatments with MJ would reduce densities of larval RWW. A second hypothesis was that MJ seed treatments would alter emergence, biomass accumulation, and yield of rice. The final hypothesis was that induction of plant resistance to the RWW would diminish as the time from seed treatment increased. In order to investigate these hypotheses, RWW densities were determined in greenhouse and field trials. Plant growth was measured in the field by assessing plant emergence, root and shoot biomass, time of heading, and yield (grain mass). Results indicated that MJ seed treatments induced resistance to RWW, although this effect decayed over time. Additionally, there were costs to plant growth and fitness; emergence and heading were delayed and biomass was reduced. Importantly, however, yields on a per-plant were not significantly reduced by MJ treatment. Overall, these results are promising and show the potential for the use of jasmonate elicitors as part of a pest management program in rice.

## Introduction

The phytohormone jasmonic acid (JA) has been identified as a primary mediator of plant responses to insect attack. Perception of herbivore feeding by a plant triggers rapid accumulation of JA at the site of attack. The biosynthesis of JA, from its precursor linolenic acid, through various active intermediates, and culminating in its conjugation with isoleucine (Ile) to form

**Funding:** MJS was supported by a grant from the Louisiana Rice Research Board and by the USDA National Institute of Food and Agriculture, Hatch project accession number 0218143. ECK was supported by a scholarship from the Louisiana Agricultural Consultants Association in 2017 and the Ray and Dorothy Young Endowed Assistantship in Agricultural Pest Management in 2018. Funders had no role in the design, data collection, analysis, decision to publish, or preparation of the manuscript.

**Competing interests:** The authors have declared that no competing interests exist.

the receptor-active JA-Ile, has been characterized [1–2]. The interaction of JA-Ile with its receptor complex in the nucleus results in a comprehensive transcriptional reorganization in the plant. Changes in gene expression result, in turn, in the local and systemic production of defense-related metabolites that cause increases in direct and indirect plant resistance to herbivores [1–3]. The well-established role of JA in induced plant defense has led to the suggestion that the JA pathway can be manipulated to protect crops from their insect pests [4–6].

One way of manipulating the JA pathway in plants is to apply JA or one of its derivatives to plants to stimulate plant resistance in anticipation of herbivore attack. Various derivatives and methods of application of JA have been tested across a range of species to investigate potential uses in integrated pest management (IPM). These prior studies include a series of experiments involving foliar applications of JA and methyl jasmonate (MJ, a methylated derivative of JA) in tomato, *Solanum lycopersicum* [5,7,8]. These studies found that treatment of plants with MJ rendered plants less suitable to insects such as *Spodoptera exigua*, likely due to an increased production of resistance-related metabolites. JA applications also increased parasitism of caterpillar pests in the field, presumably via increased emission of volatile organic compounds that attract parasitoids [5]. Another study showed that treatment of cabbage seeds, *Brassica oleracea*, with JA induced resistance to *Plutella xylostella* L. and *Myzus persicae*. Resistance was manifested by reduced growth rates and reduced population growth for caterpillars and aphids, respectively [6]. Additionally, a greenhouse study using seed treatments of MJ and JA on several crops including tomato, cowpea, soybean, and wheat, showed a reduction in nematode infection in cowpea and tomato plants [9]. These results show that the JA pathway is highly conserved across the plant kingdom and that exogenous jasmonates can elicit resistance to diverse arthropod and nematode challenges. Potential benefits of elicitors in integrated pest management programs include reductions in pest feeding injury, promotion of natural biological control by induction of volatile organic compounds, and reduced environmental contamination.

The benefits of exogenous jasmonate as a tool for pest management may, however, be counterbalanced by costs associated with their use, such as reductions in plant growth and fitness resulting from upregulation of the JA cascade. In the Muttucumaru et al. (2013) study cited above, germination and growth of tomato and wheat were not affected by jasmonate treateatment, but cowpea showed delayed germination and reduced shoot mass and soybean showed reduced germination [9]. Drenching roots of native populations of tobacco, *Nicotiana attenuata*, with MJ induced resistance to *Trimerotropis pallidipennis*, the pallid-winged grasshopper, probably due to large increases in the production of nicotine, but the plants produced significantly fewer seeds in the absence of herbivory [4]. Tomato treated with foliar applications of JA showed greater resistance to *Manduca sexta*, but increased resistance resulted in a 35% reduction in fruit production [10]. While there has been progress in understanding the hormonal crosstalk and other mechanisms responsible for tradeoffs between growth and defense in plants [11], there is still much to be learned in this area to determine whether the benefits of jasmonate application to crop plants outweigh the potential costs.

One aspect of herbivore or elicitor-induced resistance for which little information is available involves the duration of induced resistance following the initial inducing event [6,12,13]. Underwood (1998) found that resistance induced by Mexican bean beetle feeding in soybean developed within three days of beetle feeding but had completely decayed 15 days after initial injury [14]. Induced responses in white clover, *Trifolium repens*, decayed or relaxed 28 days after their induction by *Mamestra brassicae* larval feeding [13]. Another study involving marine algae showed induced responses were sustained for as long as 38 days following mechanical injury [15]. The idea that induced defenses deteriorate over time is plausible, because the most widely accepted hypothesis for the evolution of induced resistance invokes

conservation of plant resources and reduction in costs associated with producing defenses in the absence of herbivory [16]. In other words, induced responses allow a plant to produce defenses only when needed and thus plants would be expected downregulate defense responses in the absence of sustained injury.

In order to determine whether the benefits of utilizing jasmonates in pest management outweigh the costs, investigations must be made using specific crop-insect interactions under realistic (field) conditions. Rice in the southern U.S. is attacked by an array of herbivore pests, the most economically damaging of which is the rice water weevil (RWW), *Lissorhoptrus oryzophilus*. Adults of this species chiefly infest young rice plants and feed on leaf blades. Oviposition occurs in leaf sheaths in seedling and tillering rice under flooded conditions [17]. Populations of larvae, which feed on the roots of flooded rice plants, are present in high numbers only after flooding, usually peaking four to five weeks after flooding and decreasing thereafter [18]. Feeding by larvae on the roots of flooded plants reduces root biomass and ultimately causes yield loss. Yield losses can exceed 20% under heavy pressure [17]. The ability to manage early season infestations or *L. oryzophilus* would prevent economic losses and benefit growers. Use of elicitors to reduce infestations of this pest could greatly reduce reliance on insecticidal seed treatments and foliar insecticides that are currently utilized to manage this pest.

The objectives of this study were to determine if treating seeds with MJ stimulated the resistance of water-seeded rice to the RWW under greenhouse and field conditions; and, if so, to assess how long the effect lasted and whether the benefits of seed treatment were counterbalanced by negative effects on rice plant growth and yield. It was hypothesized that the MJ seed treatment would induce resistance to RWW under greenhouse conditions, and that this effect would decay over time. Additionally, it was hypothesized that treating seeds with MJ would reduce larval densities on treated plants in the field. Finally, it was hypothesized that there would be a tradeoff between plant defense and growth, represented by delays or reductions in rice plant emergence, heading, biomass and yield.

## Materials and methods

### Seed treatment

The procedures used to treat seeds were similar for the greenhouse and field experiments. To treat seeds, 160 g of unhulled seeds (cv. 'Cheniere') were soaked in 250 ml solutions of MJ in glass flasks. For greenhouse experiments, 0 mM (Control), 2.5 mM (Low), and 5.0 mM (High) solutions of MJ were used. For field experiments, 0 mM and 2.5 mM solutions of MJ were used. The solutions were prepared by mixing 250 mL of distilled water, 250 μL of Tween 20 (0.1% v:v) (Polyoxyethyle sorbitant monolaurate, Bio-Rad laboratories, Inc.) and methyl jasmonate (Sigma-Aldrich Co. St. Louis Missouri) as needed to achieve the desired final concentration. Flasks containing seeds and treatment solutions were gently shaken on an orbital platform shaker for 24 hours on a laboratory bench at room temperature. For the greenhouse experiment, seeds were sown immediately after the 24-hour soak period; for field experiments, seeds were sown approximately 24 hours after removing seeds from the MJ solutions.

### Greenhouse experiment

The duration of resistance induced by seed treatments of MJ at three concentrations (Control, 0mM; Low, 2.5 mM; and High, 5.0 mM) were investigated in a greenhouse experiment on the campus of Louisiana State University, Baton Rouge. Rice seeds were sown in a soil mix (2:1:1, topsoil: peat moss: sand) in 15cm diameter pots at the rate of five seeds/pot. The rice variety 'Cheniere', a long-grain, conventional, inbred variety, was used in all experiments. Plants were maintained in the greenhouse under ambient lighting at approximately 29.0˚C -33.0˚C and

were watered regularly to maintain adequate soil moisture. Plants in pots were thinned to two plants per pot approximately two weeks after planting. Plants from seeds sown on the same day were used for experiments at two different time points, 15 and 29 days after planting. Adult rice water weevils used to infest plants were collected from rice fields at the Louisiana State University Agricultural Center H. Rouse Caffey Rice Research Station in Crowley, Acadia Parish, Louisiana, one day prior to use. Weevils were maintained from collection until use in plastic containers with water and rice leaves. Infestation cages, which were constructed of cylindrical wire frames (46 cm diam, 61 cm height) covered with a fine mesh screening, were placed in wooden basins lined with heavy black plastic that allowed basins to be flooded. One pot from each of the three treatments (Control, Low, and High) was placed in each of the cages on the day of infestation. Separate groups of plants were infested at two time points; one set of plants was infested fifteen days after planting (early three-leaf stage), and the second set was infested twenty nine days after planting (early tillering stage). For infestations at both plant stages, adult weevils were released in cages at a density of four weevils per plant. Basins were flooded to a depth of approximately 24 cm, and weevils were allowed to feed, mate, and oviposit on plants in cages for five days. The design was a completely randomized block design in which cages were blocks and individual pots were replicates. There were ten blocks at the fifteen-day infestation and twelve at the twenty-nine infestation.

At the end of the five-day infestation period, plants were removed from cages and any adult weevils found on plants were removed. Assessment of effects of seed treatments involved counting the numbers of first instars emerging from eggs deposited on plants during the infestation period. Both plants from each pot were removed from soil, and soil was gently washed from roots. Individual plants were then placed in labeled test tubes containing distilled water. Test tubes were arranged on racks and placed in a growth chamber (28$^{\circ}$C, 14:10hr L: D). Weevils infesting plants treated in this manner eclosed from eggs, emerged from leaf sheaths, and settled on roots or on the bottoms of test tubes. First instars were counted by shaking roots free of larvae, pouring water from test tubes into a Petri dish, and visually inspecting for larvae. Plants were placed back into their respective test tubes immediately after counting and tubes were replenished with distilled water. Larvae were counted daily until no larvae were recovered for two days in a row.

## Field experiments

**Study site and rice cultivation.** Field experiments were conducted at the H. Rouse Caffey Rice Research Station (30.231422$^{\circ}$N and -92.379583$^{\circ}$W, 7m asl). The soil type at this site is a Crowley silt loam with a pH of 7.1 and 12% organic matter. Fields at this site have been in a two-year rice-fallow rotation for over 30 years. The experiment relied on natural infestations of weevils, the populations of which are consistently high at this site.

Rice in field experiments was water seeded. Water seeding of rice involves soaking dry, unhulled seeds in water (or solution of MJ) for approximately 24 hours, removing seeds from water for another 24 hours to allow germination to begin, and then casting partially germinated seed into flooded field plots. Generally, once seeds are cast into a field, floodwaters are removed to allow seedlings to establish, and permanent flood is established within a few weeks of seeding. The seeding rate for field experiments was the equivalent of 200 kg/ha. Plots measured 1m$^2$ and were seeded with 20g of treated seed.

Two field experiments were conducted, one in the 2017 growing season and one in the 2018 growing season, to determine the effects of MJ seed treatment on levels of infestation of RWW larvae and on rice growth and grain yield. Treatments in both experiments were factorial combinations of MJ seed treatment and insecticide application. The control treatment,

"Control", consisted of plots seeded with rice not treated with either MJ or insecticide. For the second treatment, "Karate", plots were not treated with MJ but were treated by making repeated applications of a pyrethroid insecticide, lambda-cyhalothrin (Karate®, Syngenta, Greensboro, NC) within 2 weeks of flooding. This resulted in plots with reduced populations of RWW, but which were not treated with MJ. For the third treatment, "MJ", the plots were sown with rice seeds treated with MJ two days prior but not treated with insecticide. This resulted in plots in which the effects of MJ on RWW infestations and rice growth and yield could be measured. In the fourth treatment, "MJ + Karate", plots were sown with rice treated with MJ and treated with repeated applications of a foliar a pyrethroid insecticide. This resulted in plots in which the effects of MJ on growth and yield could be assessed under reduced RWW pressure. Field experiments employed a randomized complete block design with four blocks and a full factorial of MJ seed treatment, and insecticide application in each block.

Insecticide applications were made to appropriate plots using Karate® at a rate of .06 kg AI/ha (Karate with Zeon Technology, Syngenta, Wilmington, Delaware). Foliar applications began one day before flooding and were repeated every four days until a total of five applications had been made to plots. Insecticide was applied using a backpack sprayer pressurized with $CO_2$ and calibrated to deliver 140 L/ha at 207 kPa through four 1002 flat fan nozzles Tee-Jet® at 51 cm spacing.

**Densities of RWW.**   In 2017 plots were permanently flooded three weeks after seeding, but in 2018, due to cool temperatures, rice growth was not sufficient to flood until nearly six weeks after seeding. The procedure used to determine densities of RWW larvae and pupae in plots after flooding involved removing root/soil core samples from each plot. Each plot was core sampled on five separate dates in 2017, from one to two months after seeding. In 2018, each plot was core sampled on two separate dates, 55 and 65 days after seeding. The tool used for removing soil/root cores was a metal cylinder with a diameter of 9.2 cm and a depth of 7.6 cm attached to a handle. This core sampler allowed for equally sized samples of soil and rice roots to be collected. Core samples were rinsed with water under moderate pressure through a metal screen bucket in order to remove soil and dislodge RWW larvae from roots. The bucket was then placed in salt water and floating larvae and pupae were counted [19].

**Plant emergence.**   Emergence of seedlings was assessed prior to flooding in both 2017 and 2018. In the 2017 experiment, emergence of rice plants was assessed 14 and 17 days after seeding by counting plants in one random quadrat (0.09m$^2$) within each plot. No pyrethroid applications had been made at this point, and for this reason seedling counts provide information only about the effects of MJ treatment. In the 2018 experiment, emergence of rice plants was assessed by taking stand counts using the same procedure used in 2017, at 12 and 22 days after seeding. Once again, no pyrethroid applications had been made at this point, and hence only the effect of MJ on plant emergence was assessed by this procedure.

**Plant biomass.**   Plant biomass was assessed after flooding in both 2017 and 2018. Biomass of roots and shoots was determined by removing whole plants from plots by hand. Plant material was rinsed thoroughly, placed in paper bags, and dried to constant weight in an oven at 60˚ C. Roots and shoots (including leaves) were separated from one another and root and shoot masses were determined using an electronic scale. Plants for biomass measurements were collected on five separate dates in 2017 and two separate dates in 2018. In 2017, collections occurred at 25, 33, 38, 45, and 52 days after seeding. Plants for biomass measurements in 2018 were collected 35 days and 49 days after seeding.

**Panicle densities.**   Panicle densities (% plants in which panicles had exerted from the flag leaf in each plot; Saichuk et al. 2014) were visually assessed in each plot on three dates in 2017 (80, 82, and 87 days after seeding) and two dates in 2018 (101 and 108 days after seeding). In

both years, the first assessment was performed when panicle emergence was first noted in control plots.

**Grain yields.** Methods for assessing treatment effects on grain yields differed between 2017 and 2018. In 2017, at grain maturity, five individual panicles were collected randomly from each plot. The panicles were threshed, and the weight of grains from the five panicles averaged for a measure of mean panicle grain mass for each plot. The resulting average panicle grain mass was used in analysis. In 2018, grain mass was measured on both a per panicle and a per plant basis. Five entire plants were collected from each plot and the number of panicles, mass of grain from each individual panicle, and total grain mass of all panicles was determined for each plant. Dates of all agricultural practices and dates of sampling are shown in S1 Table.

## Statistical analysis

All analyses were performed in SAS software version 9.4 (SAS Institute 2013). PROC GLIMMIX was used to analyze all count and continuous data as described below. Poisson distribution and log link were used for all count data. All continuous data were tested for normality with PROC UNIVARIATE and the Shapiro-Wilk test ($P < 0.05$). Biomass data for roots and shoots from both 2017 and 2018 were log transformed. Two outliers were removed from the 2017 root biomass dataset. Panicle grain mass from 2017 was log transformed. Means, standard deviations, and standard errors were determined in PROC MEANS.

For the greenhouse experiment, effects of MJ seed treatments on numbers of first instars emerging from plants were analyzed separately for the experiments done at the two plant ages. Generalized linear mixed models were used to analyze treatment effects with block as a random effect and MJ seed treatment concentration as the fixed effect. Means were tested for significant differences using Tukey's HSD ($P < 0.05$).

Data from field experiments included larval counts from core samples, counts of emerged plants, biomass of roots and shoots, % heading, and grain masses. Data sets from the 2017 and 2018 experiments were analyzed separately to take into account differences in the environment between years, but multiple samplings within years were pooled. Generalized linear mixed models using PROC GLIMMIX were used to analyze treatment effects for field data. Block was used as a random effect and fixed effects were MJ seed treatment, insecticide application, their interaction, and time. Emergence counts were the exception as this data was collected before pyrethroid applications had been made, and for this reason only the effects of MJ seed treatment were analyzed. Means were tested for significant differences using Tukey's HSD ($P < 0.05$) for all pooled datasets. In 2017, in addition to the pooled analysis, data from each core sample date and each biomass sample date were analyzed individually to determine mean, standard error, and thus effect size.

## Results

### Greenhouse experiment

Treatment of seeds with MJ significantly reduced numbers of first instars emerging from plants at both high and low concentrations relative to the control, but the effect deteriorated over time (i.e., the magnitude of the MJ effect was smaller for plants infested 29 days after seeding than for plants infested 15 days after seeding). For plants infested 15 days after seeding, both low and high concentrations of MJ significantly reduced RWW counts relative to the control ($F_{2,18} = 41.76$; $P<0.001$) (Fig 1), and counts in the High treatment were significantly lower than counts in the Low treatment. For plants infested 29 days after treating seeds, only the low concentration significantly reduced RWW densities relative to the control ($F_{2,22} =$

6.87; P = 0.005) (Fig 1). However, the reduction in counts of emerging weevils was only about half of the reduction seen for the first infestation.

### Field experiments

**Densities of RWW.** The 2017 experiment was sampled five times after flooding to evaluate effects of MJ and Karate treatments on densities of RWW larvae associated with rice roots. Pooled analysis of the data over all five core-sampling dates showed that both treatment of seeds with MJ and repeated application of Karate reduced densities of RWW immatures on roots compared to the control treatment (Fig 2). The combination of Karate applications and MJ seed treatment reduced weevil densities more effectively than MJ alone, but no more effectively than Karate alone, a fact that was reflected by a significant interaction of Karate application and MJ treatment ($F_{1,149}$ = 3.99; P = 0.048). Treatment of seeds with MJ reduced densities of RWW immatures by approximately 30% relative to controls ($F_{1,149}$ = 13.58; P<0.001) (Fig 2). Applications of the insecticide Karate significantly reduced RWW densities ($F_{1,149}$ = 76.21; P<0.001), by 51% compared to controls (Fig 2). The effect of time was also significant

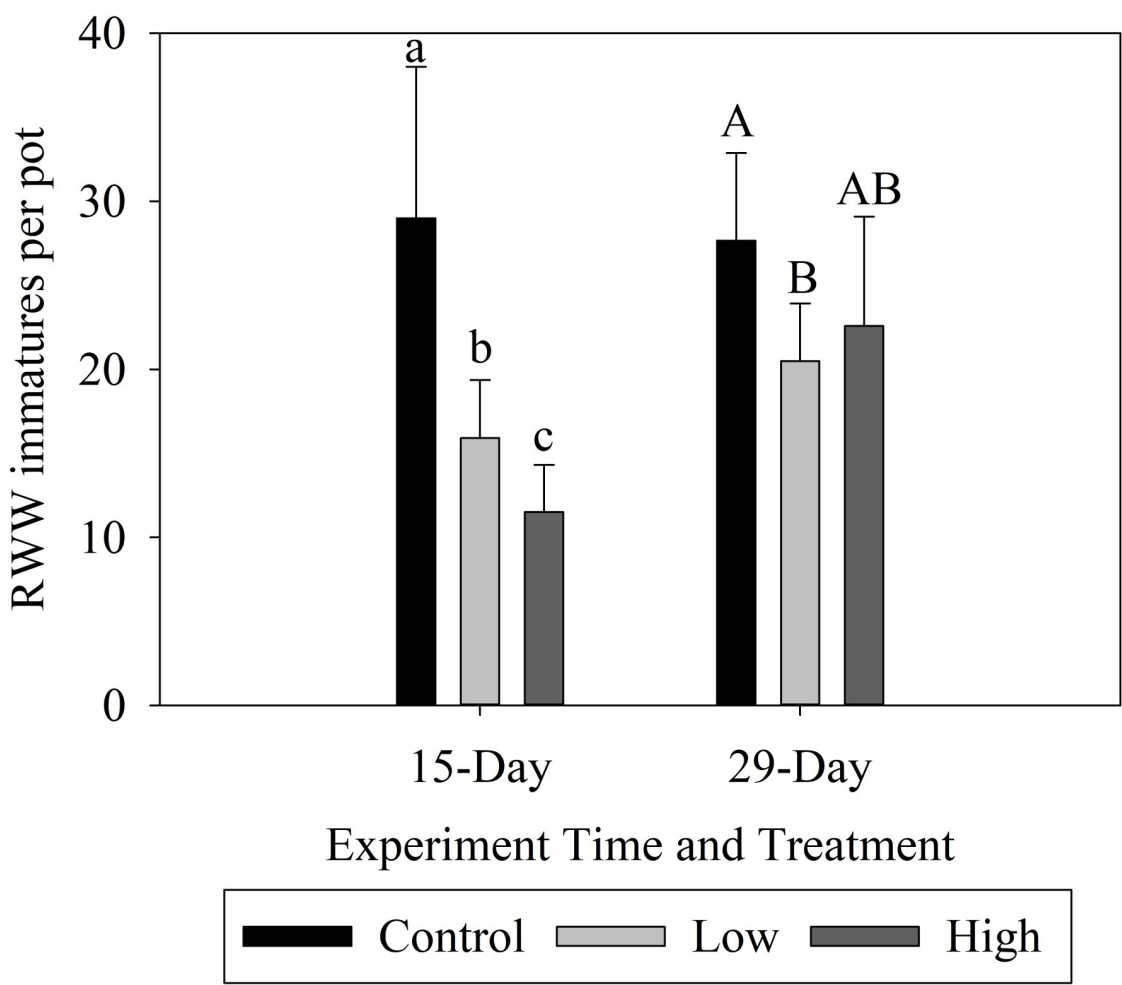

**Fig 1. Effect of seed treatment with methyl jasmonate at three concentrations on counts of first instars emerging from plants (larvae per pot ±S.E.) in a greenhouse experiment performed in 2018.** Plants were infested with adult RWW 15 days and 29 days after treatment of seeds. Bars accompanied by different letters represent means that differed significantly by Tukey's HSD test at α < .05.

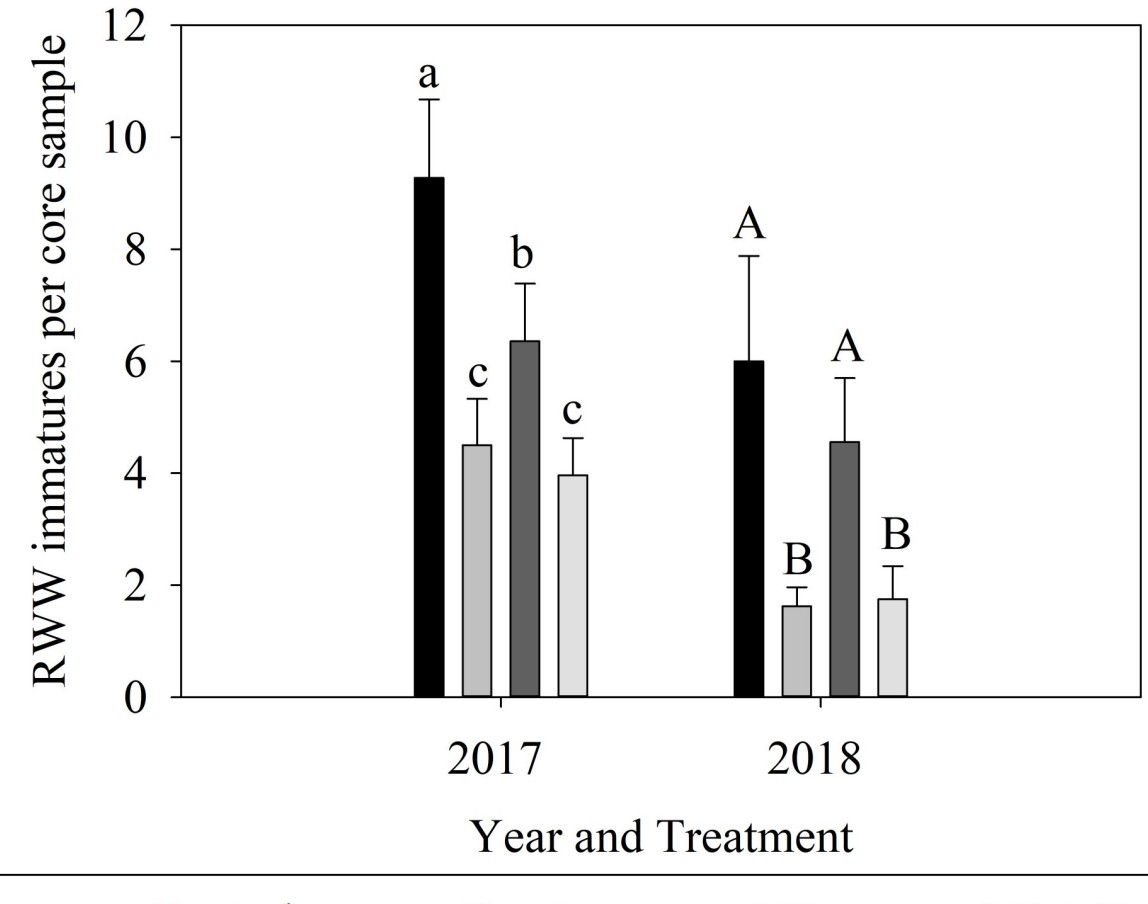

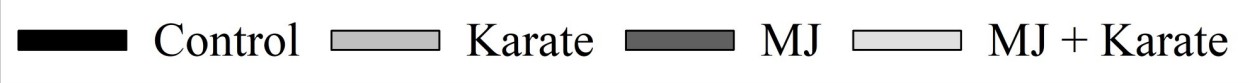

**Fig 2. Effects of factorial combinations of methyl jasmonate seed treatment (MJ) and insecticide application (Karate) on densities of RWW larvae (immature RWW per core sample ± SE) associated with roots of rice plants in field plots exposed to natural infestations of RWW in 2017 and 2018.** Plots were seeded with untreated seed (Control) or with MJ-treated seeds (MJ), sprayed four times with a pyrethroid insecticide after flooding (Karate), or seeded with MJ-treated seeds and sprayed with insecticide (MJ + Karate). Analyses in 2017 used pooled data from five core-sampling dates. Data from one core sampling date in 2018 are presented. Bars accompanied by different letters represent means that differed significantly by Tukey's HSD test at α<0.05.

($F_{4,149}$ = 71.45; P<0.001), as overall densities of RWW immatures increased over the five core sampling dates (S2 Table).

In 2018, core samples were taken 18 and 28 days after flooding. On the first sampling date, densities of RWW immatures were near zero and therefore too low to infer treatment effects. On the second sampling date, densities of RWW immatures were 73% lower in Karate-treated plots than in control plots ($F_{1,27}$ = 26.09; P<0.001) (S2 Table), but neither MJ nor the interaction of MJ and Karate had a significant effect on RWW densities ($F_{1,27}$ = 0.20; P = 0.656; $F_{1,27}$ = 0.62; P = 0.439) (Fig 2).

**Plant emergence.** In the 2017 experiment, 48% fewer plants had emerged 14 days after seeding in plots with MJ-treated rice than in plots seeded with non-treated rice (Fig 3), indicating that seedling emergence was significantly reduced or delayed by MJ treatment ($F_{1,59}$ = 198.30; P<0.001). Three days later, numbers of plants emerged were approximately 10% higher overall ($F_{1,59}$ = 5.83; P = 0.019) (S2 Table), but the effect of MJ persisted. There was no interaction between MJ and time ($F_{1,59}$ = 0.56; P = 0.456) (S2 Table).

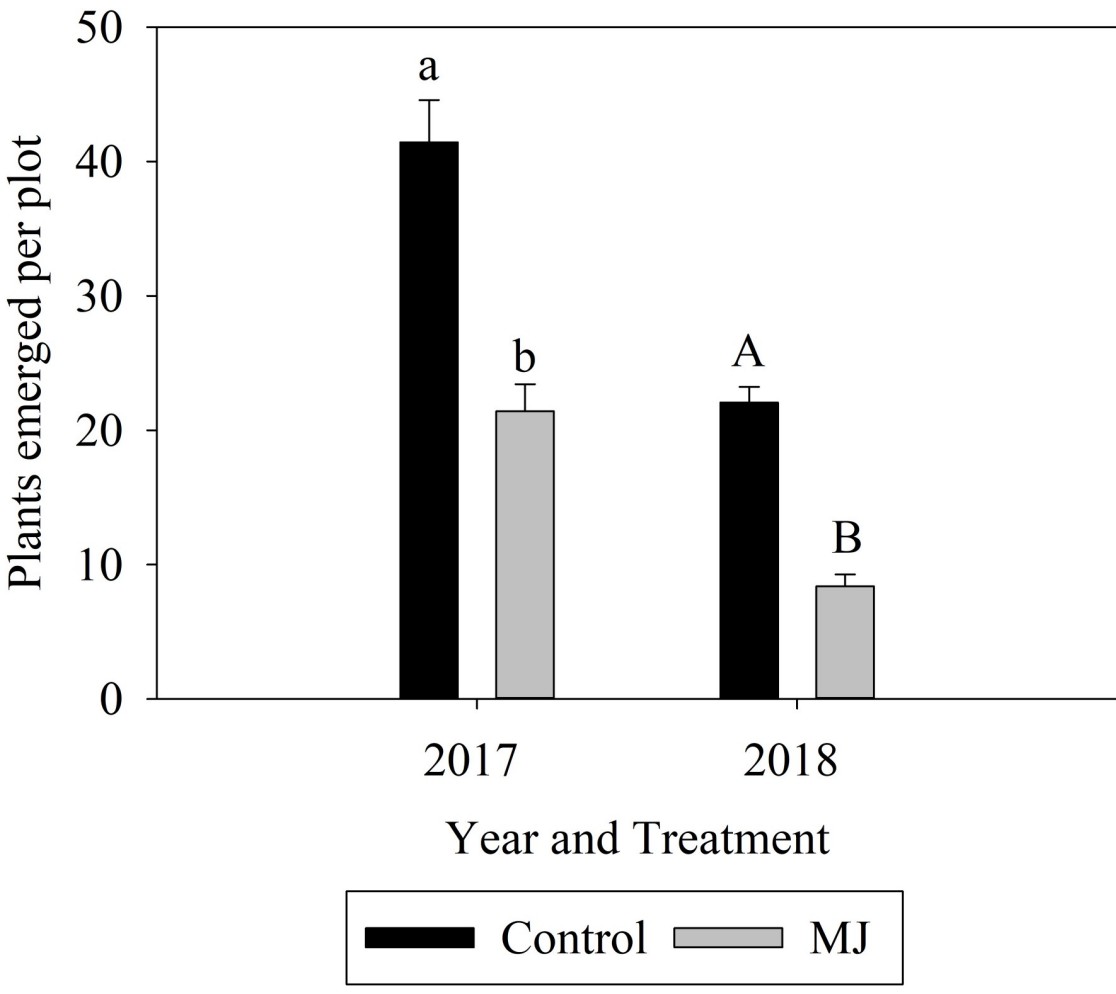

**Fig 3. Effects of treatment of rice seeds with methyl jasmonate (MJ) at a concentration of 2.5 mM on numbers of emerged rice seedlings in a 0.09m² area (numbers of plants ± SE).** Pooled data from 14 and 17 days after seeding in 2017 and 12 and 22 days after seeding in 2018 are shown. Bars accompanied by different letters represent means that differed significantly by Tukey's HSD test at α < .05.

In the 2018 experiment, plant densities increased by 22% from the first sampling date (12 days after seeding) to the second (22 days after seeding) ($F_{1,57}$ = 20.74; P<0.001) (S2 Table). Over both sampling dates, emergence was 62% lower in MJ-treated plots ($F_{1,57}$ = 184.85; P<0.001) than in control plots (Fig 3). The interaction of MJ and time was significant, but the effect of MJ was consistent across both sample dates ($F_{1,57}$ = 7.32; P = 0.009) (S2 Table).

**Plant biomass.** In the 2017 experiment, root biomass was measured at five separate time points, from 25 days after planting to 52 days after planting. Biomass of roots significantly increased over time ($F_{4,148}$ = 374.03; P<0.001) (S2 Table). Over all sampling dates, treatment of seeds with MJ reduced root biomass by 32% compared to controls ($F_{1,148}$ = 24.17; P<0.001) (Fig 4). In contrast, application of Karate increased root biomass by 4% when compared to controls. Root biomass in Karate treatments was significantly higher than root biomasses in the MJ and MJ + Karate but not the Control plots ($F_{1,148}$ = 11.05; P = 0.001) (Fig 4). The interaction between MJ and Karate was not significant for root biomass ($F_{1,148}$ = 0.00; P = 0.974).

Shoot biomass also increased over the five sampling dates ($F_{4,150}$ = 364.40; P<0.001) and was reduced by MJ seed treatment ($F_{1,150}$ = 26.97; P<0.001) (S2 Table) (Fig 4). Shoot biomass in MJ-

treated plots was 22% lower over all five sampling dates than shoot biomass in control plots (Fig 4). There was no significant effect of Karate ($F_{1,150} = 6.60$; P = 0.011) or the interaction of MJ and Karate ($F_{1,150} = 0.17$; P = 0.682) on shoot biomass (Fig 4). However, there were significant differences among treatments which show that plots treated with MJ alone (MJ) had 35% lower biomass than plots treated with Karate alone (Karate), and plots treated with MJ and Karate (MJ + Karate) showed an 18.5% reduction compared to plots treated with Karate alone (Karate) (Fig 4).

In the 2018 experiment, root and shoot biomasses were measured 35 and 49 days after seeding. Over the two sampling dates, root biomass was not significantly affected by MJ treatment ($F_{1,56} = 3.73$; P = 0.59), Karate application ($F_{1,56} = 1.01$; P = 0.319), or their interaction ($F_{1,56} = 0.23$; P = 0.634) (S2 Table). However, biomass of roots did significantly increase over the two samplings ($F_{1,56} = 598.15$; P<0.001) (S2 Table).

Shoot biomass was significantly affected by MJ in 2018 ($F_{1,56} = 4.91$; P = 0.031), but differences among individual treatments were not observed after Tukey adjustment. Shoot biomass

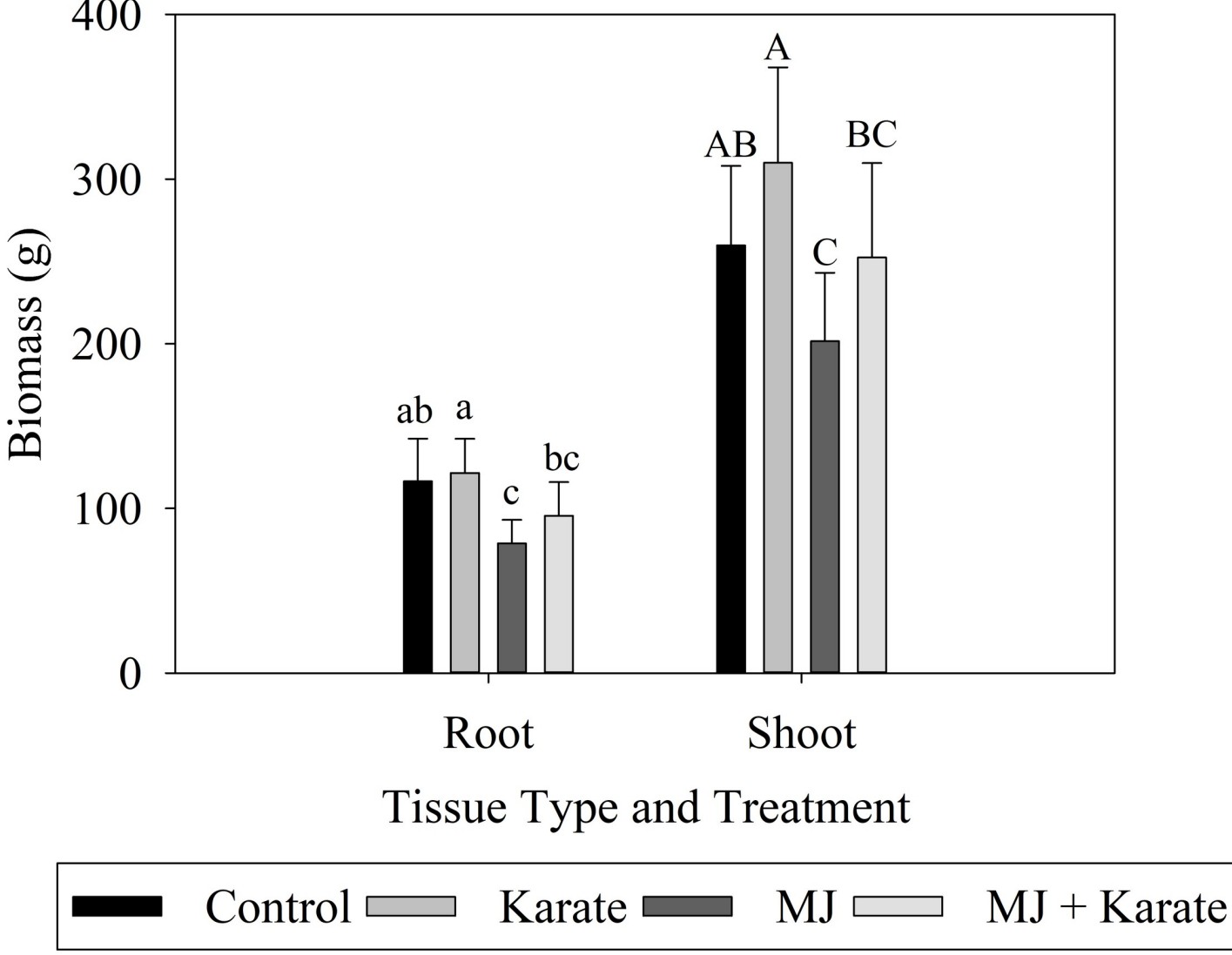

**Fig 4. Effects of factorial combinations of methyl jasmonate seed treatment (MJ) and insecticide application (Karate) on root and shoot biomasses (grams per plant ±S.E.) of rice plants over five sampling dates in 2017.** Average root (A) and shoot (B) masses of plants in each treatment: MJ–treated (MJ), insecticide-treated (Karate), MJ- and insecticide-treated (Both), and non-treated plots (Control), are shown.

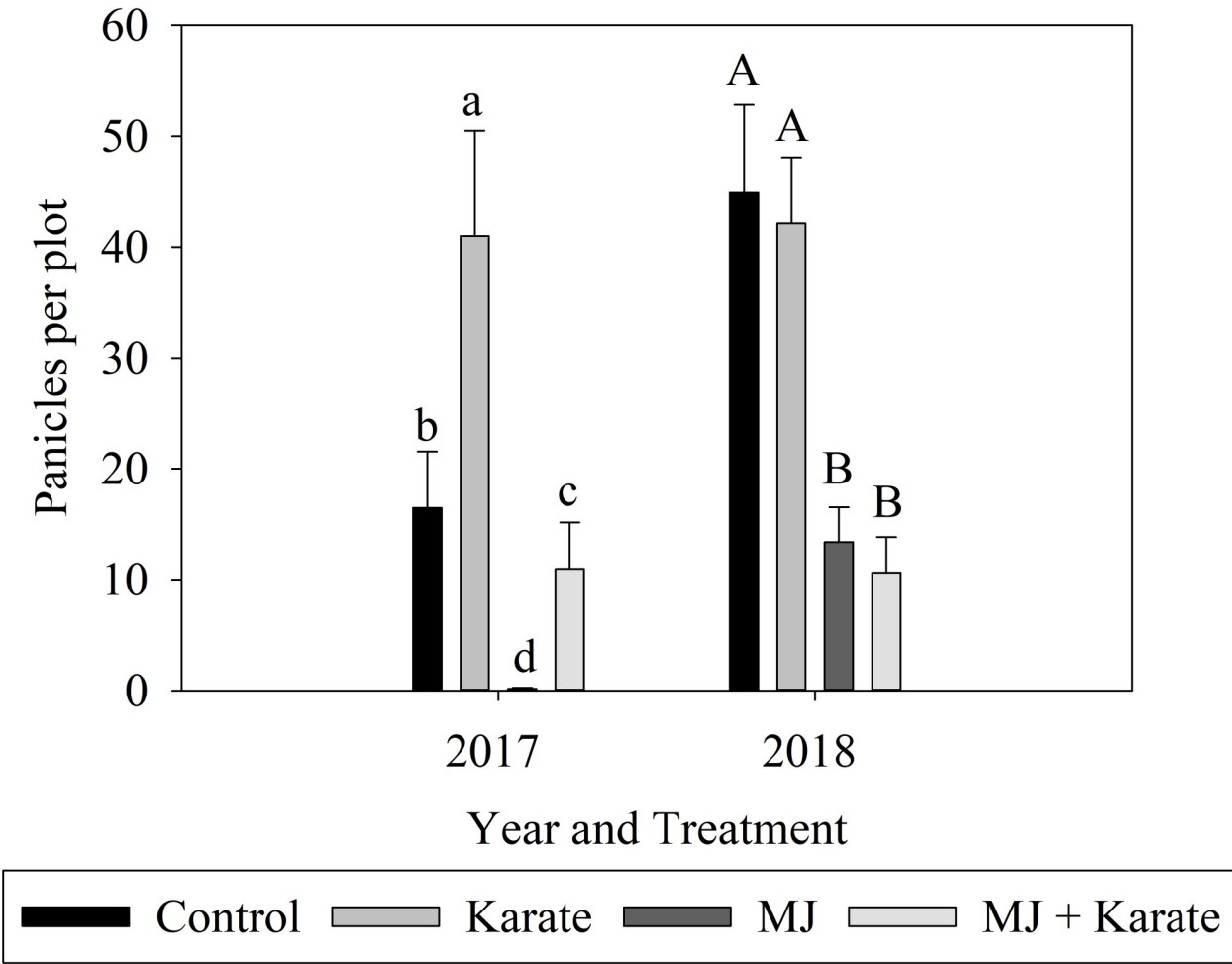

**Fig 5. Effects of factorial combinations of methyl jasmonate seed treatment (MJ) and application of insecticide (Karate) on panicle densities (numbers of panicles per plot ±S.E.).** Data were pooled over three sampling dates from 80 to 87 days after seeding in 2017 and two sampling dates from 101 to 108 days after seeding in 2018. Treatments included MJ–treated (MJ), insecticide-treated (Karate), MJ- and insecticide-treated (Both), and non-treated plots (Control). Bars accompanied by different letters represent means that differed significantly by Tukey's HSD test at α < .05.

was not affected by Karate application ($F_{1,56}$ = 0.92; P = 0.341), or by the interaction of MJ and Karate ($F_{1,56}$ = 0.03; P = 0.873) (S2 Table). Shoot biomass significantly increased as the season progressed ($F_{1,56}$ = 531.02; P<0.001) (S2 Table).

 **Panicle densities.**   Numbers of panicles that had emerged from flag leaves in plots 80, 82, and 87 days after seeding in 2017 were significantly affected by MJ treatment ($F_{1,89}$ = 138.32, P<0.001), Karate application ($F_{1,89}$ = 108.64; P<0.001) and the interaction between MJ and Karate ($F_{1,89}$ = 37.92; P<0.001)(Fig 5). Panicle densities in plots treated with MJ alone were 99% lower than in Control plots while panicle densities in plots treated with Karate alone were 60% higher than in Control plots (Fig 5). Panicle densities in plots treated with both MJ and Karate were 33% lower than controls. Panicle densities significantly increased over the three sampling dates ($F_{2,89}$ = 517.53; P<0.001) (S2 Table).

 Numbers of panicles per plots at 101 and 108 days after seeding in 2018 were significantly affected by MJ ($F_{1,56}$ = 383.78; P<0.001), but not by Karate ($F_{1,56}$ = 0.51; P = 0.477), or the interaction of MJ and Karate ($F_{1,56}$ = 1.73; P = 0.193). Panicle densities in plots treated with MJ alone and with MJ and Karate were 45% and 46% lower, respectively, than panicle densities in

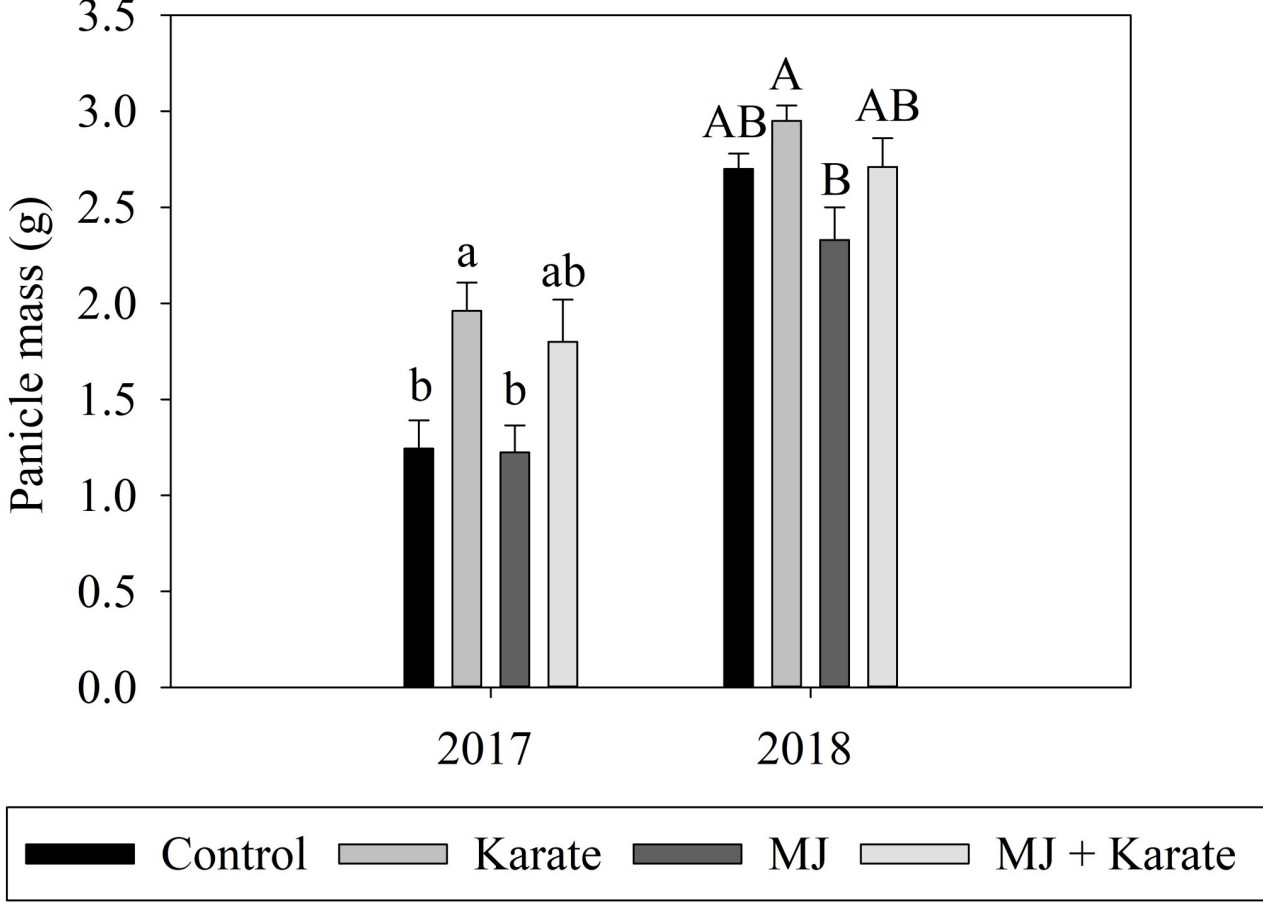

**Fig 6. Effects of factorial combinations of methyl jasmonate seed treatment (MJ) and application of insecticide (Karate) on individual panicle mass (grams of grain per panicle ±S.E.) in the experiments performed in 2017 and 2018.** Mean panicle masses for MJ–treated (MJ), insecticide-treated (Karate), insecticide and MJ-treated (Both), and non-treated plots (Control) are shown. Bars accompanied by different letters represent means that differed significantly by Tukey's HSD test at α < .050.

control plots (Fig 5). Panicle densities were significantly higher on the second sampling date ($F_{1,56}$ = 1168.96; P<0.001) than on the first (S2 Table).

**Grain yields.** Results from 2017 showed no effect of MJ ($F_{1,24}$ = 0.21; P = 0.655) on grain mass per panicle, and the interaction of MJ and Karate was also not significant ($F_{1,24}$ = 0.15; P = 0.702). However, mean grain mass per panicle was 37% higher from Karate-treated plots (Karate) than in control plots (Control) ($F_{1,24}$ = 15.73; P = 0.001) (Fig 6).

In 2018, there were significant effects of both MJ seed treatment and Karate on grain mass per panicle (MJ: $F_{1,27}$ = 5.75; P = 0.024) (Karate: $F_{1,27}$ = 6.07; P = 0.020) (Fig 6). Panicle grain masses from plots treated with MJ only (MJ) were 13.7% lower than controls (Control) and 21% lower than grain masses from Karate-treated plots, but only the latter difference was significant (Fig 6). The interaction was not significant (MJ*Karate $F_{1,27}$ = 0.23; P = 0.633).

In addition to assessing grain mass from individual panicles, entire plants were assessed for total number of panicles and total grain mass per plant in 2018. The average number of panicles per plant was not significantly affected by MJ or Karate treatment or their interaction (MJ: $F_{1,25}$ = 0.82; P = 0.373) (Karate: $F_{1,25}$ = 0.62; P = 0.439)(Interaction: $F_{1,25}$ = 0.18; P = 0.675) and total grain mass per plant was not affected by MJ ($F_{1,27}$ = 0.44; P = 0.512), Karate ($F_{1,27}$ = 0.57; P = 0.456), or their interaction ($F_{1,27}$ = 0.00; P = 0.991)(S2 Table).

## Discussion

Progressive elucidation over the past three decades of the signaling pathways that mediate plant responses to herbivory [20–21] has led to suggestions that these pathways could be manipulated in crop plants to manage pests. Numerous studies have shown that it is possible to stimulate plant resistance by applying jasmonates (JA or its derivatives) to plants. However, these experiments have mostly been conducted under semi-controlled conditions, and several questions remain to be fully addressed before elicitors of plant resistance can be incorporated into management programs for insect pests in crops.

One important question that has not been adequately addressed is whether the resistance induced by jasmonates is strong and durable enough under realistic field conditions. Based on prior research in both the greenhouse and the field [22–23], it was hypothesized that treating seeds with MJ would induce resistance to RWW under field conditions. Results from the greenhouse and from the 2017 field experiment supported this hypothesis. Reductions in RWW densities achieved with seed treatments in the field in 2017 (~30%) were not lower than reductions achieved by applications of a broad-spectrum insecticide made within two weeks of flooding (~50%), but four applications of insecticides were needed to achieve this level of control. Results in 2017 were the first-field level experiments testing MJ seed treatments as a pest management tool in rice.

The results in 2018 differed somewhat from results in 2017. No significant differences in RWW densities due to seed treatments with MJ were observed, although RWW densities were 24% lower in plots treated with MJ. Plant growth in 2018 was delayed due to cold weather, which necessitated a delay in flooding. This delay increased the amount of time between the seed treatments and RWW oviposition, which is triggered by flooding. It is likely that the effects of induction by MJ seed treatment in this experiment had decayed by the time RWW were ovipositing in the field. This interpretation is supported by our greenhouse experiments. In the greenhouse, MJ-induced resistance deteriorated by 30 days post-seed treatment. Most studies of induced resistance investigate responses immediately or nearly immediately following imposition of stress or application of elicitors, and relatively few studies have investigated the duration of induced responses, an aspect vital to the use of induced resistance in IPM. The current study demonstrates that MJ-induced resistance in rice is not maintained throughout the season, a fact consistent with hypotheses regarding tradeoffs of growth and defense and the evolution of induced resistance [16].

Karate applications were more effective at reducing densities of RWW than was MJ treatment, but it is important to note from an IPM perspective that plots in the Karate treatment received four to five applications of the insecticide, whereas plots in the MJ treatment received a single exposure to MJ at the seed stage. Another result to consider is that the combination of MJ and Karate can produce a synergistic effect such as the one seen on a single core sampling date in 2017 experiment. On this date, RWW densities in plots that received both MJ and Karate (Both) were about half as high as RWW densities in control plots, whereas MJ or Karate alone reduced densities by 14% or 25%, respectively. However, on five of the six core sampling dates there was no evidence of synergism, and on two of these dates the RWW densities in plots that received both MJ and Karate applications were higher than densities in the plots that recycled Karate applications only.

Practical use of jasmonates in crop IPM will also require an application method that is economically feasible and compatible with accepted grower practices. In this study, rice was water-seeded, a method long used for rice production in southwest Louisiana and a method compatible with soaking seeds before planting. Soaking seeds in a solution of MJ might also be compatible with the method commonly used for rice production in much of Asia. Rice in this

region is planted densely in seedbeds via broadcast seeding. When seedlings reach two to three weeks of age they are transplanted into fields. In this case, rice seedlings may be protected by MJ treatment in the seedbed and during the first few weeks after transplanting. However, most rice in the southern U.S and elsewhere is drill-seeded into a dry seedbed, and in those case use of MJ seed treatments would require development of new methods to coat seeds with jasmonates in which seeds remain dry.

Perhaps the most crucial issue involved in the use of jasmonates in pest management is their potential impact on plant growth and yield. In this study, there were clear effects of MJ seed treatment on emergence and growth of plants. MJ seed treatment significantly delayed or reduced germination in both 2017 and 2018. In the pooled analyses of 2017 data, MJ treatment also reduced early season root and shoot biomass when compared to the control plots. However, plants appeared to recover such that effects on biomass were not significant later in the season. Plots in which the RWW feeding pressure was lessened by application of insecticides (Karate and MJ+Karate), likely provide the best information on the effects of MJ on plant growth. In 2017, both root and shoot biomasses were significantly lower in the MJ+Karate plots than in the Karate-only treatment on the first two sample dates but not later suggesting that plants recover biomass loss from the hormone treatment. There were no significant effects on root or shoot biomass in 2018.

In addition to the effects of MJ on plant growth, heading (% of plants in which panicles had exerted from the flag leaf in each plot) was significantly delayed in both the 2017 and 2018 experiments in MJ-treated (MJ and MJ+Karate) plots compared to plots not treated with MJ (Control and Karate treatments). These delays may have been residual effects from the delay in emergence. Plants sometimes tolerate herbivory by altering flowering or fruiting phenology and extending their growing season [24]. In a review on plant tolerance to herbivory Tiffin (2000) provides dozens of examples of plants postponing development in order to tolerate herbivory [25]. However, in the 2018 experiment, numbers of panicles per plant at grain maturity were no lower in MJ-treated plots than in plots not treated with MJ, again showing that the primary effect of MJ may have been to delay development of panicles rather than reduce panicle numbers.

Despite the delays in heading, evidence that MJ seed treatments reduced yields was mixed. In both 2017 and 2018, yields (masses of grains per panicle) in plots treated with MJ only were no higher than yields from plots in the Control treatment despite the reductions in weevil densities observed in MJ plots. However, yields were higher from plots in which Karate was applied (Karate treatment) than from Control plots. These two results suggest that gains in yield resulting from suppression of weevils were counterbalanced by the costs of MJ treatment. On the other hand, when weevils were suppressed by Karate applications and MJ was applied (Both treatment), yields were not significantly lower than when Karate was applied by itself (Karate). This suggests that MJ treatment did not necessarily reduce yields and that one reason for decreased yields in the MJ-only treatment was the fact that weevil suppression in these plots was not as high as weevil suppression in Karate treatments. Disentangling the yield costs and benefits of MJ treatment in rice will require additional experiments in which MJ seed treatments are made in the absence of weevils. Growth-defense tradeoffs in relation to responses induced by MJ seed treatments have previously demonstrated in tomato (Paudel et al. 2014). In this study, increased activity of the resistance-related enzyme polyphenol oxidase was noted in plants treated as seeds with MJ, but this increase in plant resistance was accompanied by reductions in plant germination, height, and fruit mass [26].

Understanding tradeoffs between plant growth and defense is critical to evaluating the potential for jasmonate treatments to provide sustainable long-term solutions for pest management in rice and other crops. From an IPM perspective, one critical feature of the experiments

reported here is that a single treatment of rice seeds with MJ was capable of causing reductions in weevil densities comparable to reductions produced by multiple applications of a broad-spectrum pyrethroid, so the potential of MJ seed treatments as an IPM tool is clear. In rice, experiments in which RWW are absent are needed to better characterize the costs of MJ induction by seed treatments. Additionally, further investigations into the effects of jasmonates on germination and heading are needed in order to determine if these aspects of plant growth are reduced or merely delayed by MJ treatments. Once these issues have been resolved, work will also be needed to evaluate the cost effectiveness of this tactic and determine its compatibility with common production practices.

## Supporting information

**S1 Table. Dates of agricultural practices.** Dates of agricultural practices including seed treatments, water seeding, flooding, and foliar sprays. Sample dates for emergence, biomass, rice water weevil core samples, heading, and harvest.
(PDF)

**S2 Table. Supplemental data.** Mean and standard error are provided for each treatment, on each sample date, for all response variables in experiments in 2017 and 2018.
(PDF)

## Acknowledgments

This manuscript was approved for publication by the Director of the Louisiana Agricultural Experiment Station, manuscript number 2019. We are grateful to the Crowley Rice Research Station staff and student employees for assistance with field experiments.

## Author Contributions

**Conceptualization:** Emily C. Kraus, Michael J. Stout.

**Data curation:** Emily C. Kraus.

**Formal analysis:** Emily C. Kraus.

**Funding acquisition:** Emily C. Kraus, Michael J. Stout.

**Investigation:** Emily C. Kraus.

**Methodology:** Emily C. Kraus, Michael J. Stout.

**Project administration:** Emily C. Kraus.

**Resources:** Michael J. Stout.

**Supervision:** Michael J. Stout.

**Visualization:** Emily C. Kraus.

**Writing – original draft:** Emily C. Kraus.

**Writing – review & editing:** Emily C. Kraus, Michael J. Stout.

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
