## [Decision Letter · Decision Letter 0]

6 Aug 2019

PONE-D-19-14964

Seed treatment using methyl jasmonate induces resistance to an herbivore but reduces plant growth in rice

PLOS ONE

Dear Dr Kraus,

Thank you for submitting your manuscript to PLOS ONE. After careful consideration, we feel that it has merit but does not fully meet PLOS ONE’s publication criteria as it currently stands. Therefore, we invite you to submit a revised version of the manuscript that addresses the points raised during the review process.

I apologize for the delay in this response. It was quite difficult to find reviewers for this paper. Having now received two reviews, I am happy to indicate that the paper only requires minor edits. Please address those issues indicated by the reviewer and we will gladly accept the paper.

We would appreciate receiving your revised manuscript by Sep 20 2019 11:59PM. To enhance the reproducibility of your results, we recommend that if applicable you deposit your laboratory protocols in protocols.io, where a protocol can be assigned its own identifier (DOI) such that it can be cited independently in the future. For instructions see: http://journals.plos.org/plosone/s/submission-guidelines#loc-laboratory-protocols

We look forward to receiving your revised manuscript.

Kind regards,

Sean Michael Prager, Ph.D.

Academic Editor

PLOS ONE

Journal Requirements:

1. We note that you have included the phrase “data not shown” in your manuscript. Unfortunately, this does not meet our data sharing requirements. PLOS does not permit references to inaccessible data. We require that authors provide all relevant data within the paper, Supporting Information files, or in an acceptable, public repository. Please add a citation to support this phrase or upload the data that corresponds with these findings to a stable repository (such as Figshare or Dryad) and provide and URLs, DOIs, or accession numbers that may be used to access these data. Or, if the data are not a core part of the research being presented in your study, we ask that you remove the phrase that refers to these data.

Reviewers' comments:

Reviewer's Responses to Questions

**Comments to the Author**

1. Is the manuscript technically sound, and do the data support the conclusions?

Reviewer #1: Yes

Reviewer #2: Yes

2. Has the statistical analysis been performed appropriately and rigorously? 

Reviewer #1: Yes

Reviewer #2: Yes

3. Have the authors made all data underlying the findings in their manuscript fully available?

Reviewer #1: Yes

Reviewer #2: Yes

4. Is the manuscript presented in an intelligible fashion and written in standard English?

Reviewer #1: Yes

Reviewer #2: Yes

5. Review Comments to the Author

Reviewer #1: This manuscript reported the effect of seed treatment with MJ on rice resistance to RWW and on plant growth. The authors found that MJ treatment of rice seeds enhanced plant resistance to larval RWW but reduced plant growth. However, the authors also observed that yields on a per-plant were not significantly reduced by MJ treatment.The work is interesting and show the potential for the use of jasmonate elicitors as part of a pest management program in rice. Overall, the experiments in the manuscript were well designed and the article was well written.

I have just some minor concerns:

Line 301, “The effect of time was also significant”, there is no data to prove this. May be it can be put in the supplementary meterials.

Line 315, “On the second sampling date, densities of RWW immatures were 73% lower in Karate treated plots than in control plots”. There was also no data to support this.

Line 322-325, “Three days later, numbers of plants emerged were approximately 10% higher overall” “There was no interaction between MJ and time”. No data support.

Line 331-332,334-335, 338-339, 351-352, 367-368, 376-377, 389-390, 407-412, no data support.

Line 355-358,365-367, no data support. Is it possible to put some supplemental tables or figures?

Reviewer #2: In this work, the authors studied the effects of methyl jasmonate (MJ) seed treatment on plant resistant to the rice water weevil (RWW) and plant growth in rice under greenhouse and field conditions. It was demonstrated that MJ treatment to seeds enhanced rice resistant to RRW but with a cost, as seedling emergence and heading were delayed and rice biomass was reduced. However, and importantly, they found the yields were not significantly impaired by the MJ treatment. Especially, this work provides useful data of effect of MJ seed treatment in field conditions for two years, and the data include investigations on the effect of MJ treatment, insecticide Karate application, and both, and the duration of MJ effects was also determined. This study is interesting and expands our understanding on the potential application of MJ to seeds as part of a pest management program.

I have only a minor suggestion: In the title, “an herbivore” could be replaced by “rice water weevil”.

6. PLOS authors have the option to publish the peer review history of their article (what does this mean?). If published, this will include your full peer review and any attached files.

Reviewer #1: No

Reviewer #2: No

---

## [Author Response · Author response to Decision Letter 0]

6 Sep 2019

The reviewers comments were mainly addressed by the additional of a supplemental table including all means and standard errors from each sample date.

5. Review Comments to the Author

I have just some minor concerns:

Line 301, “The effect of time was also significant”, there is no data to prove this. May be it can be put in the supplementary materials.

A supplementary table was created which includes the mean and standard error for each treatment on each individual sampling date. (Line 303)

Line 315, “On the second sampling date, densities of RWW immatures were 73% lower in Karate treated plots than in control plots”. There was also no data to support this.

A supplementary table was created which includes the mean and standard error for each treatment on each individual sampling date. (Line 317)

Line 322-325, “Three days later, numbers of plants emerged were approximately 10% higher overall” “There was no interaction between MJ and time”. No data support.

A supplementary table was created which includes the mean and standard error for each treatment on each individual sampling date. (Lines 325-326)

Line 331-332,334-335, 338-339, 351-352, 367-368, 376-377, 389-390, 407-412, no data support. Line 355-358,365-367, no data support. Is it possible to put some supplemental tables or figures?

Figure 4 contains much of this data, but was not noted in the text. This has been corrected. Additionally a supplementary table was created which includes the mean and standard error for each treatment on each individual sampling date. (Lines 333-334, 337, 341, 345, 354, 356, 357, 361, 365-366, 370-371, 380,393,415)

Reviewer #2: 

I have only a minor suggestion: In the title, “an herbivore” could be replaced by “rice water weevil”.

This suggestion was included and the title altered

---

## [Editor Report · Decision Letter 1]

9 Sep 2019

Seed treatment using methyl jasmonate induces resistance to rice water weevil but reduces plant growth in rice

PONE-D-19-14964R1

Dear Dr. Kraus,

We are pleased to inform you that your manuscript has been judged scientifically suitable for publication and will be formally accepted for publication once it complies with all outstanding technical requirements.

With kind regards,

Sean Michael Prager, Ph.D.

Academic Editor

PLOS ONE
---

## [Editor Report · Acceptance letter]

16 Sep 2019

PONE-D-19-14964R1 

Seed treatment using methyl jasmonate induces resistance to rice water weevil but reduces plant growth in rice 

Dear Dr. Kraus:

I am pleased to inform you that your manuscript has been deemed suitable for publication in PLOS ONE. Congratulations! Your manuscript is now with our production department. 

With kind regards,

on behalf of

Dr. Sean Michael Prager 

Academic Editor

PLOS ONE